# Expression of Trefoil Factor 1 (TFF1) in Cancer: A Tissue Microarray Study Involving 18,878 Tumors

**DOI:** 10.3390/diagnostics14192157

**Published:** 2024-09-28

**Authors:** Florian Lutz, Soo-Young Han, Seyma Büyücek, Katharina Möller, Florian Viehweger, Ria Schlichter, Anne Menz, Andreas M. Luebke, Ahmed Abdulwahab Bawahab, Viktor Reiswich, Martina Kluth, Claudia Hube-Magg, Andrea Hinsch, Sören Weidemann, Maximilian Lennartz, David Dum, Christian Bernreuther, Patrick Lebok, Guido Sauter, Andreas H. Marx, Ronald Simon, Till Krech, Christoph Fraune, Natalia Gorbokon, Eike Burandt, Sarah Minner, Stefan Steurer, Till S. Clauditz, Frank Jacobsen

**Affiliations:** 1Institute of Pathology, University Medical Center Hamburg-Eppendorf, 20251 Hamburg, Germany; f.lutz@uke.de (F.L.); sooyoung.han@icloud.com (S.-Y.H.); s.bueyuecek@uke.de (S.B.); ka.moeller@uke.de (K.M.); f.viehweger@uke.de (F.V.); r.schlichter@uke.de (R.S.); a.menz@uke.de (A.M.); luebke@uke.de (A.M.L.); v.reiswich@uke.de (V.R.); m.kluth@uke.de (M.K.); c.hube@uke.de (C.H.-M.); a.hinsch@uke.de (A.H.); s.weidemann@uke.de (S.W.); m.lennartz@uke.de (M.L.); d.dum@uke.de (D.D.); c.bernreuther@uke.de (C.B.); p.lebok@uke.de (P.L.); g.sauter@uke.de (G.S.); t.krech@uke.de (T.K.); c.fraune@uke.de (C.F.); n.gorbokon@uke.de (N.G.); e.burandt@uke.de (E.B.); s.minner@uke.de (S.M.); s.steurer@uke.de (S.S.); t.clauditz@uke.de (T.S.C.); f.jacobsen@uke.de (F.J.); 2Department of Basic Medical Sciences, College of Medicine, University of Jeddah, Jeddah 21589, Saudi Arabia; bawahab2002@hotmail.com; 3Institute of Pathology, Clinical Center Osnabrueck, 49078 Osnabrueck, Germany; 4Department of Pathology, Academic Hospital Fuerth, 90766 Fuerth, Germany; andreas.marx@klinikum-fuerth.de

**Keywords:** TFF1, tissue microarray, immunohistochemistry, human cancers

## Abstract

**Background/Objectives:** Trefoil factor 1 (TFF1) plays a role in the mucus barrier. **Methods:** To evaluate the prevalence of TFF1 expression in cancer, a tissue microarray containing 18,878 samples from 149 tumor types and 608 samples of 76 normal tissue types was analyzed through immunohistochemistry (IHC). **Results:** TFF1 staining was detectable in 65 of 149 tumor categories. The highest rates of TFF1 positivity were found in mucinous ovarian carcinomas (76.2%), colorectal adenomas and adenocarcinomas (47.1–75%), breast neoplasms (up to 72.9%), bilio-pancreatic adenocarcinomas (42.1–62.5%), gastro-esophageal adenocarcinomas (40.4–50.0%), neuroendocrine neoplasms (up to 45.5%), cervical adenocarcinomas (39.1%), and urothelial neoplasms (up to 24.3%). High TFF1 expression was related to a low grade of malignancy in non-invasive urothelial carcinomas of the bladder (*p* = 0.0225), low grade of malignancy (*p* = 0.0003), estrogen and progesterone receptor expression (*p* < 0.0001), non-triple negativity (*p* = 0.0005) in invasive breast cancer of no special type, and right-sided tumor location (*p* = 0.0021) in colorectal adenocarcinomas. **Conclusions:** TFF1 IHC has only limited utility for the discrimination of different tumor entities given its expression in many tumor entities. The link between TFF1 expression and parameters of malignancy argues for a relevant biological role of TFF1 in cancer. TFF1 may represent a suitable therapeutic target due to its expression in only a few normal cell types.

## 1. Introduction

Trefoil factor 1 (TFF1/pS2) belongs to one of three highly conserved proteins of the trefoil factor family, which are co-expressed with mucins in the gastrointestinal tract. TFF1 is predominantly expressed in gastric mucosa [1], where it plays a role in protection and healing of the mucus barrier (summarized in [2]). Therefore, TFF1 expression is stimulated in response to acute mucosal injury and chronic inflammation (summarized in [3]). In superficial cells of gastric mucosa, TFF1 is co-expressed and bound to Mucin 5AC (MUC5AC), supporting a role for TFF1 in the packaging and function of the gastric mucosa [4]. Based on the identification of recurrent TFF1 deletions, missense mutations, and hypermethylation in gastric cancer, TFF1 has been considered as a tumor suppressor protein (summarized in [5]). In line with this notion, TFF1-deficient mice serve as a popular model for gastric cancer development [6]. All TFF1-deficient mice develop adenomas in the gastric mucosa, and 30% of them eventually progress to invasive carcinomas [7].

Because the TFF1 expression in normal tissue strongly predominates in stomach mucosa, TFF1 immunohistochemistry (IHC) has been suggested as a marker for gastric cancer [8]. However, studies have shown that not all gastric adenocarcinomas produce TFF1 and that various other tumor entities can express TFF1. In several cancer types, elevated TFF1 expression has been linked to either poor [9,10] or improved patient prognosis [11,12,13,14]. However, the reported frequencies of TFF1 positivity vary considerably for most tumor types. For example, the range of reported TFF1-positive cases ranged from 16% to 90% in gastric cancer [13,15,16,17], from 51% to 98% in cholangiocarcinoma [18,19,20], from 0% to 100% in breast cancer [14,21,22,23], from 0% to 100% in colorectal cancer [24,25,26,27], from 0% to 40% in pulmonary adenocarcinoma [28,29,30,31], from 0% to 14% in serous ovarian carcinoma [28,32,33], from 0% to 25% in clear cell renal cell carcinoma [28,32,34], from 0% to 91% in prostate cancer [28,35,36], and from 55% to 95% in pancreatic cancer [10,37,38]. The use of different antibodies, staining protocols, and criteria to determine TFF1 positivity represents the most likely cause for these controversial data.

To better understand the role of TFF1 protein expression in cancer and to elucidate the potential diagnostic role of TFF1 IHC, an extensive survey of TFF1 immunostaining in a broad range of tumor types and under highly standardized conditions is needed. TFF1 expression was, thus, evaluated in more than 18,000 tumor tissue samples from 149 different tumor types and subtypes as well as 76 different non-neoplastic tissue types by IHC in a tissue microarray (TMA) format in this study.

## 2. Materials and Methods

### 2.1. Tissue Microarrays (TMAs)

The normal TMA included 8 samples from 8 different donors from 76 different normal tissue types (608 samples on one slide). The cancer TMAs contained 18,878 primary tumor probes from 149 tumor entities and tumor subtypes. Data on histopathological and molecular parameters were available for cancers of the colorectum (*n* = 2351), stomach (*n* = 327), pancreas (*n* = 598), urinary bladder (*n* = 1073), and the breast (*n* = 2139). A detailed description of the composition of both normal and cancer TMAs is available in the Results section. All tumor probes were from the archives of the Institute of Pathology, University Hospital of Hamburg, Germany; the Department of Pathology, Academic Hospital Fuerth, Germany; and the Institute of Pathology, Clinical Center Osnabrueck, Germany. A detailed description of the TMA manufacturing process is available in previous publications [39,40]. Per patient/tumor, one tissue spot (diameter: 0.6 mm) was used. The preparation of TMAs from archived diagnostic tissue remnants and their analysis for research purposes as well as the analysis of patient data were approved by local laws (HmbKHG, §12) and the local ethics committee (Ethics Committee Hamburg, WF-049/09). All work was conducted in accordance with the Declaration of Helsinki. Immunohistochemical data on MUC5AC expression were available from a previous study [41].

### 2.2. Immunohistochemistry (IHC)

Freshly cut TMA sections were immunostained in one experiment on one day. TMA sections were deparaffinized with xylol, rehydrated using a graded alcohol series, and incubated in an autoclave at 121 °C for 5 min in Dako Target Retrieval Solution, pH9 (Agilent Technologies, Santa Clara, CA, USA; #S2367) for heat-induced antigen retrieval. For blocking of endogenous peroxidase activity, TMA sections were incubated in Dako REAL Peroxidase-Blocking Solution (Agilent Technologies, Santa Clara, CA, USA; #S2023) for 10 min. Primary antibody specific for TFF1 (mouse monoclonal, MSVA-482M, MS Validated Antibodies, Hamburg, Germany; #5784-482M) was applied at 37 °C for 60 min at a dilution of 1:150. For antibody validation, the TMA with normal tissue probes was analyzed by the rabbit recombinant monoclonal TFF1 antibody [EPR3972] (Abcam; Cambridge, UK; #ab92377, dilution 1:900) under identical protocol. Bound antibody was visualized with Dako REAL EnVision Detection System Peroxidase/DAB+, Rabbit/Mouse kit (Agilent Technologies, Santa Clara, CA, USA; #K5007) according to the manufacturer’s directions. The sections were counterstained with hemalaun. For all tumor tissues, the staining intensity was recorded (semi-quantitatively as 0, 1+, 2+, 3+), and the fraction of positive tumor cells was estimated as percentage. For statistical analyses, four groups were determined: negative, tumors without any staining; weak, tumors with 1+ staining intensity in ≤70% of neoplastic cells and 2+ intensity in ≤30% of neoplastic cells; moderate, tumors with 1+ staining intensity in >70% of neoplastic cells, 2+ intensity in 31–70%, or 3+ intensity in ≤30% of neoplastic cells; strong, tumors with 2+ intensity in >70% or 3+ intensity in >30% of neoplastic cells.

### 2.3. Statistics

Statistical calculations were performed with JMP^®^ software (Version 17) SAS^®^, Cary, NC, USA). Contingency tables, the chi^2^-test, and Fisher’s exact test were performed to search for associations between TFF1 immunostaining and tumor phenotype and MUC5AC immunostaining.

## 3. Results

### 3.1. Technical Issues

A total of 16,817 (89.1%) of 18,878 tumor samples were interpretable in our TMA analysis. Non-interpretable samples demonstrated the absence of unequivocal tumor cells or a complete lack of individual tissue spots. A sufficient number of samples of each normal tissue type was evaluable (≥4).

### 3.2. TFF1 in Normal Tissues

TFF1 immunostaining was always cytoplasmic. It was strongest in superficial epithelial cells of the stomach, while the deeper glandular cells remained TFF1 negative. Scattered goblet cells with a significant TFF1 positivity were also seen in the small intestine and the colorectum. A weak to moderate TFF1 immunostaining was also observed in some goblet cells of the respiratory epithelium; groups of mucinous glandular cells in bronchial, sublingual, and submandibular glands; and in a fraction of luminal breast epithelial cells. Occasionally, a few cells with a distinct TFF1 positivity were also observed in the urothelium (mostly umbrella cells) or in gall bladder epithelial cells. Representative images are shown in Figure 1.

All these staining patterns were seen by both MSVA-482M and EPR3972 (Appendix A). TFF1 staining was absent in skeletal muscle, heart muscle, smooth muscle, myometrium of the uterus, corpus spongiosum of the penis, ovarian stroma, fat, skin (including hair follicle and sebaceous glands), surface epithelium of the tonsil, ectocervix, squamous epithelium of the esophagus, decidua, placenta, bone marrow; lymph node, spleen, thymus, tonsil, liver, pancreas, Brunner gland of the duodenum, cortex and medulla of the kidney, prostate, seminal vesicle, testis, epididymis, lung, endocervix, endometrium, fallopian tube, ovary (including corpus luteum and theca and granulosa cells of follicular cysts) adrenal medulla, thyroid, parathyroid gland, cerebellum, cerebrum and the pituitary gland.

### 3.3. TFF1 in Tumor Tissues

TFF1 immunostaining was detectable in 3347 (19.9%) of the 16,817 analyzable tumors, including 2093 (12.4%) with weak, 740 (4.4%) with moderate, and 514 (3.1%) with strong positivity. Overall, 65 (43.6%) of 149 tumor categories showed detectable TFF1 expression, with 32 (21.5%) tumor categories containing at least one case with strong positivity (Table 1).

Representative images of TFF1-positive tumors are shown in Figure 2. The highest rate of TFF1 positivity was found in mucinous carcinomas of the ovary (76.2%), colorectal adenomas and adenocarcinomas (47.1–75.0%), neoplasms of the breast (11.8–72.9%), bilio-pancreatic adenocarcinomas (42.1–62.5%), gastro-esophageal adenocarcinomas (40.4–50.0%), neuroendocrine neoplasms of various sites of origin (7.1–45.5%), adenocarcinomas of the cervix (39.1%), and in urothelial neoplasms (5.5–24.3%).

The ranking order of all TFF1-positive entities is shown in Figure 3. The relationship between TFF1 expression and clinically important histopathological and molecular tumor features in colorectal, gastric, pancreatic, urinary bladder, and breast cancer is shown in Table 2.

High TFF1 expression was significantly associated with a low grade of malignancy in non-invasive (pTa) urothelial carcinomas of the urinary bladder (*p* = 0.0072), low grade of malignancy (*p* < 0.0001), estrogen (ER) and progesterone receptor (PR) expression and non-triple negativity (*p* < 0.0001 each) in invasive breast cancer of no special type, as well as with right-sided tumor location (*p* = 0.0021), MMR deficiency (*p* = 0.0292), RAS mutations (*p* < 0.0001), and BRAF mutation (*p* = 0.0331) in colorectal adenocarcinomas. The extent of TFF1 staining was unrelated to histopathological parameters of malignancy in 312 gastric and 485 pancreatic adenocarcinomas.

### 3.4. Comparison with MUC5AC

Data on MUC5AC expression were available for 7870 of our tumors, which were evaluated for TFF1. A total of 50 entities showed expression of either TFF1 or MUC5AC. There was a tendency towards a co-expression of TFF1 and MUC5AC, but this relationship was rather weak. Among 7870 tumor samples with available TFF1 and MUC5AC data, 573 (7.3%) showed expression of both TFF1 and MUC5AC, 1391 (17.7%) showed expression of only TFF1, 325 (4.1%) showed expression of only MUC5AC, and 5581 (70.9%) showed expression of none of the two proteins (*p* < 0.0001). MUC5AC/TFF1 co-expression was particularly common in mucinous carcinoma of the ovary, gastro-intestinal, and bilio-pancreatic neoplasms but also occurred in other entities. A graphical representation of the expression of MUC5AC and TFF1 in 50 tumor entities is given in Figure 4.

## 4. Discussion

The successful analysis of 16,817 tumors from 149 tumor types and subtypes resulted in a comprehensive overview on the expression of TFF1 in human tumors. The ranking order of 65 tumor entities according to their rate of TFF1 positivity is a key result of our study (Figure 3). The comparative representation of our data and previously published data (Figure 5) demonstrates that this information could not be retrieved from literature databases as many tumor entities have, so far, not been analyzed, and data were often highly discordant for entities that have been analyzed in multiple studies. Our data show that TFF1 positivity occurs most commonly in mucinous carcinomas of the ovary, colorectal adenocarcinomas, breast cancer, bilio-pancreatic, and gastro-esophageal adenocarcinomas, neuroendocrine neoplasms, adenocarcinomas of the cervix uteri, and in urothelial neoplasms. Considering that 65 different tumor entities contained TFF1-positive cases and that the rate of positivity did not exceed 80% in any of the most frequently positive cancer entities, we do not believe that TFF1 IHC has a relevant role for the distinction of cancer entities. However, TFF1 IHC may be useful to select patients for potential future therapies using, for example, antibody drug conjugates. Only recently, the de novo expression of TFF1 has been associated with gemcitabine resistance in pancreatic adenocarcinomas, and TFF1 silencing increased sensitivity to gemcitabine in vitro and in vivo [42].

The striking discrepancy between TFF1 expression in only a few normal tissues and TFF1 expression in so many different cancer types demonstrates that TFF1 upregulation is a common phenomenon in cancer, which may have a significant functional role. The increase in TFF1 staining that was seen in colon adenomas as compared to normal colon mucosa further supports this notion. Earlier studies confirmed TFF1 expression in colon adenomas [8,43,44] and also reported TFF1 staining from other preneoplastic lesions, including, for example, intestinal metaplasia [45]. An oncogenic role would be most conceivable for a protein that is often overexpressed in cancer as compared to its cell of origin. In line with this notion, several authors have indeed described evidence for the oncogenic effects of TFF1. For example, TFF1 promoted anchorage-independent growth in colon carcinoma cells [43], phosphoinositide 3-kinases (PI3K) dependent migration and invasion in gastric carcinoma cells [46], and cyclooxygenase-2 (COX-2) and EGF receptor (EGFR)-dependent angiogenesis in kidney and colonic cancer cells [47]. The molecular mechanism underlying a putative oncogenic function is largely unknown. Rodrigues et al. [43] reported that TFF1 upregulates the phosphatase CDC25A, one of the most crucial cell cycle regulators and activator of CDKs [48], suggesting that both factors cooperate with other oncogenic pathways to drive a adenoma–carcinoma transition in colon carcinoma. However, the significant associations found between TFF1 expression and several favorable tumor features in bladder and breast cancer in combination with the absence of significant links with unfavorable prognostic parameters in colorectal, gastric, and pancreatic cancers at least argues against a significant role of TFF1 for driving cancer aggressiveness. TFF1 RNA expression data derived from The Cancer Genome Atlas (TCGA) database also pinpoint a favorable prognosis for breast and endometrium cancer with high TFF1 expression (https://www.proteinatlas.org/ENSG00000160182-TFF1/pathology, accessed on 9 January 2024). In line with these findings, several functional studies have rather supported a tumor-suppressive than oncogenic role for TFF1. For example, the overexpression of TFF1 suppressed nuclear factor kappa-light-chain-enhancer of activated B-cells (NF-kB) signaling and, therefore, pro-tumorigenic and metastatic properties in colon and gastric cancer cells [49,50] as well as ß-catenin signaling in hepatocellular carcinoma cells [51]. TFF1 knockdown enhanced anchorage-independent growth in breast cancer cells in vivo and in vitro [52], reduced apoptosis and induced proliferation in gastric cancer cells [53], and accelerated the development of esophageal adenocarcinoma [54] and hepatocellular carcinoma in vivo [51].

Previous studies analyzing the prognostic role of TFF1 have provided rather controversial results. For breast cancer, gastric adenocarcinoma and pancreatic ampullary adenocarcinoma studies have found associations between high TFF1 expression levels and both favorable [12,13,14,17,55,56] and unfavorable tumor features [10,57,58]. Significant associations of high TFF1 expression with favorable tumor features or favorable prognosis were also found in gallbladder adenocarcinoma [11], while high TFF1 expression was linked to unfavorable tumor parameters or poor patient outcomes in adenocarcinoma of the lung [30,31], colorectal adenocarcinoma [9,59], and in urothelial carcinoma [60]. Another study in renal cell carcinomas did not suggest any relationship between TFF1 expression and parameters of cancer aggressiveness [34].

The availability of data from an earlier study enabled us to compare our TFF1 results with MUC5AC data [41]. Although these proteins are frequently co-expressed and act cooperatively in normal tissues, the relationship between the expression of TFF1 and MUC5AC was rather weak in cancer, since only 7% of 2289 cancer samples with TFF1 and/or MUC5AC expression expressed both proteins. We, therefore, assume that MUC5AC and TFF1 are functionally less dependent on each other in neoplastic than in normal tissues. That mucinous carcinoma of the ovary, gastro-intestinal, and bilio-pancreatic neoplasms showed co-expression of TFF1 and MUC5AC (20% to 53%) more commonly than other tumor entities is in line with some previous studies describing high rates of MUC5AC/TFF1 co-expression in neoplasms of the colon, liver, stomach, and gallbladder [8,18,61,62]. However, other authors found lower co-expression rates in gastric [63] and colorectal adenocarcinomas [25], and markedly higher frequencies of co-expression were reported for cancers from the esophagus, lung, and bladder [31,64,65].

To validate our assay, our IHC results in normal tissue were compared with RNA results from three different publicly available databases [66,67,68,69] and with immunohistochemical data from an independent anti-TFF1 antibody. The antibody comparison study included 76 different categories of normal tissues to ensure the broadest possible range of expressed proteins. The assay validity was supported by the detection of TFF1 strong immunostaining in the stomach, which was the only normal organ for which RNA expression had previously been reported. The additional positive TFF1 stainings in scattered goblet cells in the small intestine, colorectum, and respiratory epithelium, some mucinous glandular cells in salivary and in bronchial glands, a fraction of luminal breast epithelial cells as well as in some cells of the urothelium and gallbladder epithelium were all validated by identical stainings seen by the independent antibody EPR3972. Given that these TFF1-positive cells constituted very small subpopulations of the respective organs, we assume that TFF1 RNA had not been detected due to a massive dilution if RNAs from total organs were analyzed.

## 5. Conclusions

Our data provide a comprehensive overview on the expression of TFF1 in human cancer. Given that TFF1 is expressed in a broad range of tumor entities, TFF1 IHC may have only limited utility for the discrimination of different tumor entities. Upregulation in multiple tumor entities and the significant link between TFF1 expression and parameters of malignancy in several tumors argue for a relevant biological role of TFF1 in cancer, however. TFF1 may also represent a suitable therapeutic target.

## Figures and Tables

**Figure 1 diagnostics-14-02157-f001:**
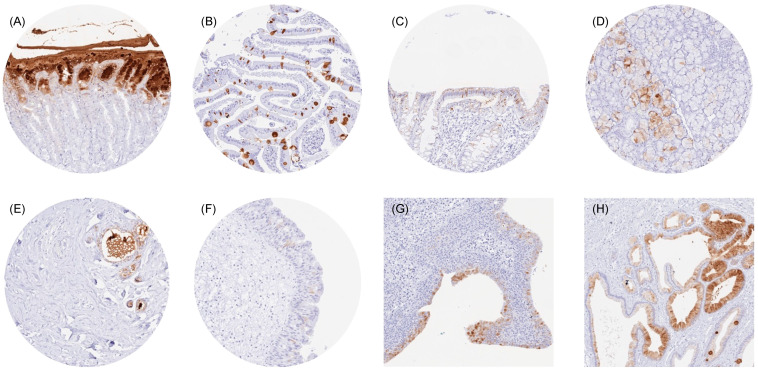
TFF1 immunostaining of normal tissues. The panels show a cytoplasmic staining of surface epithelial cells but not of glands in the stomach (**A**), subsets of goblet cells in the duodenum (**B**) and the colon (**C**), a subset of mucinous cells in the submandibulary gland (**D**), a subset of luminal epithelial cells and intraluminal mucus in the breast (**E**), a small subset of urothelial cells (mostly umbrella cells) in the renal pelvis (**F**), a large subset of urothelial cells in an inflamed urinary bladder (**G**), and (focally) in epithelial cells of the gallbladder (**H**).

**Figure 2 diagnostics-14-02157-f002:**
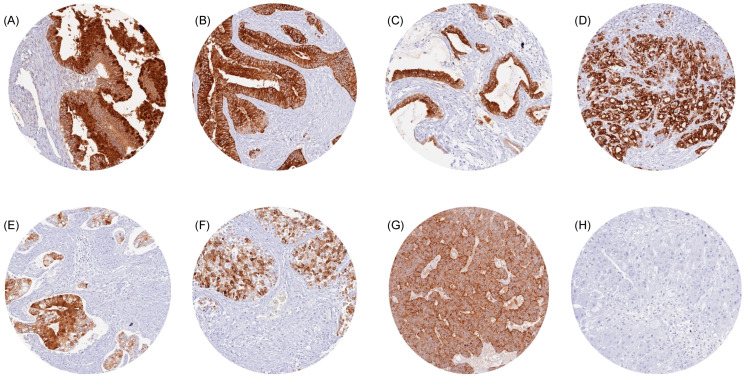
TFF1 immunostaining in cancer. The panels show TFF1 positivity in a mucinous carcinoma of the ovary (**A**), a colorectal adenocarcinoma (**B**), a pancreatic adenocarcinoma (**C**), a gastric adenocarcinoma (**D**), an adenocarcinoma of the cervix (**E**), urothelial carcinoma of the bladder (**F**), and a neuroendocrine tumor of the lung (**G**). TFF1 staining is absent in a hepatocellular carcinoma (**H**).

**Figure 3 diagnostics-14-02157-f003:**
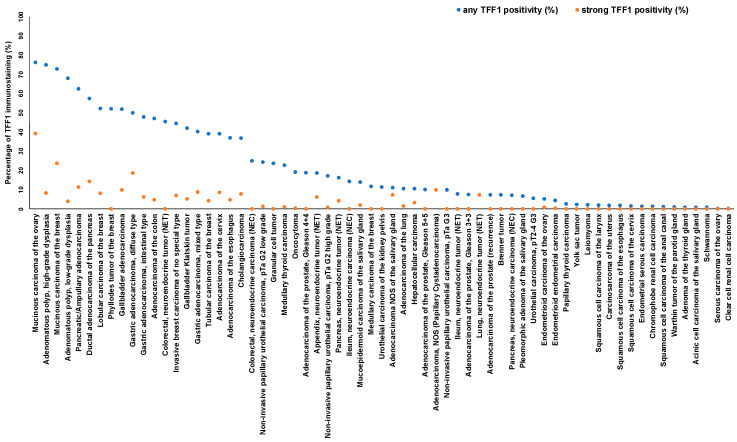
Ranking order of TFF1 immunostaining in tumors. Both the percentage of positive cases (blue dots) and the percentage of strongly positive cases (orange dots) are shown.

**Figure 4 diagnostics-14-02157-f004:**
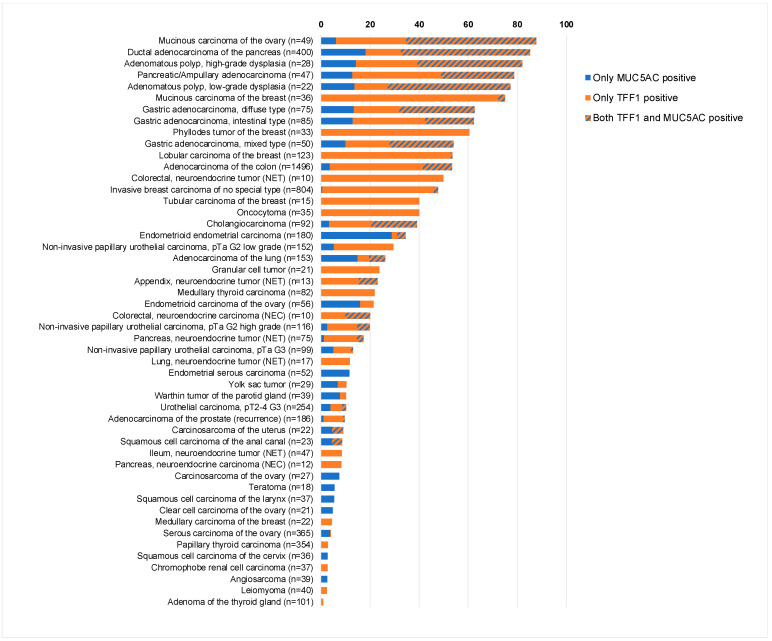
Comparison of TFF1 and MUC5AC immunostaining.

**Figure 5 diagnostics-14-02157-f005:**
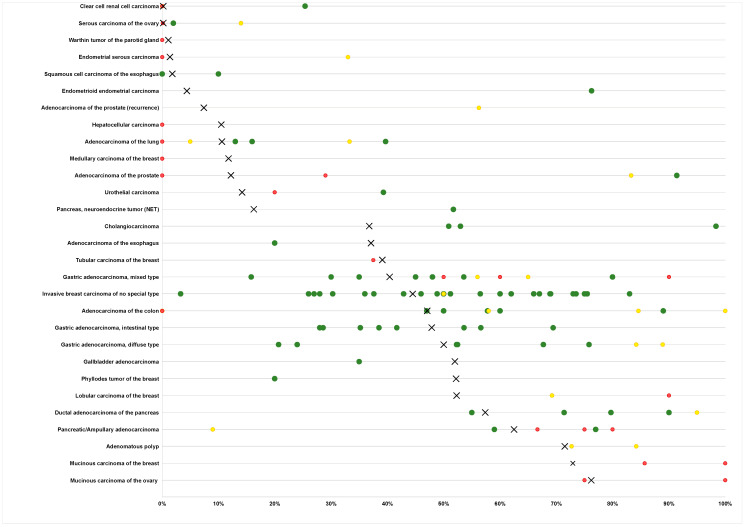
Comparison with previous TFF1 literature. An “X” indicates the fraction of TFF1 positive cancers in the present study, dots indicate the reported frequencies from the literature for comparison: red dots mark studies with ≤10 analyzed tumors, yellow dots mark studies with 11 to 25 analyzed tumors, and green dots mark studies with >25 analyzed tumors.

**Table 1 diagnostics-14-02157-t001:** TFF1 immunostaining in human tumors: neg. = negative, mod. = moderate, str. = strong, ca. = carcinoma.

			TFF1 Immunostaining				
Tumor Category	Tumor Entity	on TMA (n)	Analy-Zable (n)	Neg. (%)	Weak (%)	Mod. (%)	Str. (%)
Tumors of the skin	Basal cell carcinoma of the skin	89	81	100.0	0.0	0.0	0.0
	Benign nevus	29	21	100.0	0.0	0.0	0.0
	Squamous cell carcinoma of the skin	145	132	100.0	0.0	0.0	0.0
	Malignant melanoma	65	58	100.0	0.0	0.0	0.0
	Malignant melanoma lymph node metastasis	86	86	100.0	0.0	0.0	0.0
	Merkel cell carcinoma	2	2	100.0	0.0	0.0	0.0
Tumors of the head and neck	Squamous cell carcinoma of the larynx	109	98	98.0	2.0	0.0	0.0
	Squamous cell ca. of the pharynx	60	60	100.0	0.0	0.0	0.0
	Oral squamous cell carcinoma	130	128	100.0	0.0	0.0	0.0
	Pleomorphic adenoma of the parotid gland	50	38	100.0	0.0	0.0	0.0
	Warthin tumor of the parotid gland	104	91	98.9	1.1	0.0	0.0
	Adenocarcinoma, NOS (Papillary Cystadenocarcinoma)	14	10	90.0	0.0	0.0	10.0
	Salivary duct carcinoma	15	10	100.0	0.0	0.0	0.0
	Acinic cell carcinoma of the salivary gland	181	110	99.1	0.0	0.9	0.0
	Adenocarcinoma NOS of the salivary gland	109	54	88.9	1.9	1.9	7.4
	Adenoid cystic carcinoma of the salivary gland	180	61	100.0	0.0	0.0	0.0
	Basal cell adenocarcinoma of the salivary gland	25	19	100.0	0.0	0.0	0.0
	Basal cell adenoma of the salivary gland	101	60	100.0	0.0	0.0	0.0
	Epithelial-myoepithelial carcinoma of the salivary gland	53	44	100.0	0.0	0.0	0.0
	Mucoepidermoid carcinoma of the salivary gland	343	250	86.0	8.0	4.0	2.0
	Myoepithelial carcinoma of the salivary gland	21	15	100.0	0.0	0.0	0.0
	Myoepithelioma of the salivary gland	11	9	100.0	0.0	0.0	0.0
	Oncocytic carcinoma of the salivary gland	12	6	100.0	0.0	0.0	0.0
	Polymorphous adenoca. low grade, of the salivary gland	41	15	100.0	0.0	0.0	0.0
	Pleomorphic adenoma of the salivary gland	53	30	93.3	6.7	0.0	0.0
Tumors of the lung, pleura and thymus	Adenocarcinoma of the lung	196	188	89.4	6.9	2.1	1.6
	Squamous cell carcinoma of the lung	80	74	100.0	0.0	0.0	0.0
	Mesothelioma, epithelioid	40	34	100.0	0.0	0.0	0.0
	Mesothelioma, biphasic	29	29	100.0	0.0	0.0	0.0
	Thymoma	29	24	100.0	0.0	0.0	0.0
	Lung, neuroendocrine tumor (NET)	29	27	92.6	0.0	0.0	7.4
Tumors of the female genital tract	Squamous cell carcinoma of the vagina	78	71	100.0	0.0	0.0	0.0
	Squamous cell carcinoma of the vulva	157	144	100.0	0.0	0.0	0.0
	Squamous cell carcinoma of the cervix	136	129	98.4	1.6	0.0	0.0
	Adenocarcinoma of the cervix	23	23	60.9	17.4	13.0	8.7
	Endometrioid endometrial carcinoma	338	295	95.6	3.7	0.3	0.3
	Endometrial serous carcinoma	86	73	98.6	1.4	0.0	0.0
	Carcinosarcoma of the uterus	57	53	98.1	0.0	1.9	0.0
	Endometrial carcinoma, high grade, G3	13	8	100.0	0.0	0.0	0.0
	Endometrial clear cell carcinoma	9	6	100.0	0.0	0.0	0.0
	Endometrioid carcinoma of the ovary	130	115	94.8	4.3	0.0	0.9
	Serous carcinoma of the ovary	580	507	99.8	0.2	0.0	0.0
	Mucinous carcinoma of the ovary	101	84	23.8	21.4	15.5	39.3
	Clear cell carcinoma of the ovary	51	45	100.0	0.0	0.0	0.0
	Carcinosarcoma of the ovary	47	47	100.0	0.0	0.0	0.0
	Granulosa cell tumor of the ovary	44	42	100.0	0.0	0.0	0.0
	Leydig cell tumor of the ovary	4	4	100.0	0.0	0.0	0.0
	Sertoli cell tumor of the ovary	1	1	100.0	0.0	0.0	0.0
	Sertoli Leydig cell tumor of the ovary	3	3	100.0	0.0	0.0	0.0
	Steroid cell tumor of the ovary	3	3	100.0	0.0	0.0	0.0
	Brenner tumor	41	41	92.7	7.3	0.0	0.0
Tumors of the breast	Invasive breast carcinoma of no special type	1764	1692	55.5	26.6	10.9	7.0
	Lobular carcinoma of the breast	363	346	47.7	27.2	17.1	8.1
	Medullary carcinoma of the breast	34	34	88.2	2.9	8.8	0.0
	Tubular carcinoma of the breast	29	23	60.9	21.7	13.0	4.3
	Mucinous carcinoma of the breast	65	59	27.1	35.6	13.6	23.7
	Phyllodes tumor of the breast	50	46	47.8	30.4	21.7	0.0
Tumors of the digestive system	Adenomatous polyp, low-grade dysplasia	50	50	32.0	52.0	12.0	4.0
	Adenomatous polyp, high-grade dysplasia	50	48	25.0	50.0	16.7	8.3
	Adenocarcinoma of the colon	2483	2256	52.9	35.3	7.0	4.7
	Gastric adenocarcinoma, diffuse type	215	176	50.0	13.6	17.6	18.8
	Gastric adenocarcinoma, intestinal type	215	192	52.1	33.9	7.8	6.3
	Gastric adenocarcinoma, mixed type	62	57	59.6	22.8	8.8	8.8
	Adenocarcinoma of the esophagus	83	62	62.9	24.2	8.1	4.8
	Squamous cell carcinoma of the esophagus	76	55	98.2	0.0	1.8	0.0
	Squamous cell carcinoma of the anal canal	91	85	98.8	0.0	1.2	0.0
	Cholangiocarcinoma	121	114	63.2	21.1	7.9	7.9
	Gallbladder adenocarcinoma	51	50	48.0	22.0	20.0	10.0
	Gallbladder Klatskin tumor	42	38	57.9	21.1	15.8	5.3
	Hepatocellular carcinoma	312	304	89.5	5.3	2.0	3.3
	Ductal adenocarcinoma of the pancreas	659	613	42.6	28.1	15.0	14.4
	Pancreatic/Ampullary adenocarcinoma	98	96	37.5	32.3	18.8	11.5
	Acinar cell carcinoma of the pancreas	18	18	100.0	0.0	0.0	0.0
	Gastrointestinal stromal tumor (GIST)	62	59	100.0	0.0	0.0	0.0
	Appendix, neuroendocrine tumor (NET)	25	16	81.3	6.3	6.3	6.3
	Colorectal, neuroendocrine tumor (NET)	12	11	54.5	18.2	27.3	0.0
	Ileum, neuroendocrine tumor (NET)	53	51	92.2	5.9	2.0	0.0
	Pancreas, neuroendocrine tumor (NET)	101	92	83.7	7.6	4.3	4.3
	Colorectal, neuroendocrine carcinoma (NEC)	14	12	75.0	16.7	8.3	0.0
	Ileum, neuroendocrine carcinoma (NEC)	8	7	85.7	0.0	14.3	0.0
	Gallbladder, neuroendocrine carcinoma (NEC)	4	4	100.0	0.0	0.0	0.0
	Pancreas, neuroendocrine carcinoma (NEC)	14	14	92.9	0.0	7.1	0.0
Tumors of the urinary system	Non-invasive papillary urothelial ca. pTa G2 low grade	177	152	75.7	19.7	3.3	1.3
	Non-invasive papillary urothelial ca., pTa G2 high grade	141	116	82.8	13.8	2.6	0.9
	Non-invasive papillary urothelial carcinoma, pTa G3	219	131	90.1	7.6	2.3	0.0
	Urothelial carcinoma, pT2-4 G3	735	600	94.5	4.0	1.5	0.0
	Squamous cell carcinoma of the bladder	22	22	100.0	0.0	0.0	0.0
	Small cell neuroendocrine carcinoma of the bladder	5	5	100.0	0.0	0.0	0.0
	Sarcomatoid urothelial carcinoma	25	23	100.0	0.0	0.0	0.0
	Urothelial carcinoma of the kidney pelvis	62	61	88.5	11.5	0.0	0.0
	Clear cell renal cell carcinoma	1286	1230	99.8	0.2	0.0	0.0
	Papillary renal cell carcinoma	368	327	100.0	0.0	0.0	0.0
	Clear cell (tubulo) papillary renal cell carcinoma	26	23	100.0	0.0	0.0	0.0
	Chromophobe renal cell carcinoma	170	151	98.7	1.3	0.0	0.0
	Oncocytoma	257	230	80.9	9.6	9.1	0.4
Tumors of the male genital organs	Adenocarcinoma of the prostate, Gleason 3 + 3	83	79	92.4	6.3	1.3	0.0
	Adenocarcinoma of the prostate, Gleason 4 + 4	80	69	81.2	14.5	4.3	0.0
	Adenocarcinoma of the prostate, Gleason 5 + 5	85	79	89.9	8.9	1.3	0.0
	Adenocarcinoma of the prostate (recurrence)	258	216	92.6	6.0	1.4	0.0
	Small cell neuroendocrine carcinoma of the prostate	2	2	100.0	0.0	0.0	0.0
	Seminoma	682	586	100.0	0.0	0.0	0.0
	Embryonal carcinoma of the testis	54	41	100.0	0.0	0.0	0.0
	Leydig cell tumor of the testis	31	31	100.0	0.0	0.0	0.0
	Sertoli cell tumor of the testis	2	2	100.0	0.0	0.0	0.0
	Sex cord stromal tumor of the testis	1	1	100.0	0.0	0.0	0.0
	Spermatocytic tumor of the testis	1	1	100.0	0.0	0.0	0.0
	Yolk sac tumor	53	42	97.6	2.4	0.0	0.0
	Teratoma	53	39	100.0	0.0	0.0	0.0
	Squamous cell carcinoma of the penis	92	90	100.0	0.0	0.0	0.0
Tumors of endocrine organs	Adenoma of the thyroid gland	113	108	99.1	0.9	0.0	0.0
	Papillary thyroid carcinoma	391	374	97.3	2.7	0.0	0.0
	Follicular thyroid carcinoma	154	146	100.0	0.0	0.0	0.0
	Medullary thyroid carcinoma	111	101	77.2	19.8	2.0	1.0
	Parathyroid gland adenoma	43	42	100.0	0.0	0.0	0.0
	Anaplastic thyroid carcinoma	45	42	100.0	0.0	0.0	0.0
	Adrenal cortical adenoma	50	37	100.0	0.0	0.0	0.0
	Adrenal cortical carcinoma	28	28	100.0	0.0	0.0	0.0
	Pheochromocytoma	50	50	100.0	0.0	0.0	0.0
Tumors of haemotopoetic and lymphoid tissues	Hodgkin’s lymphoma	103	90	100.0	0.0	0.0	0.0
	Small lymphocytic lymphoma, B-cell type (B-SLL/B-CLL)	50	50	100.0	0.0	0.0	0.0
	Diffuse large B cell lymphoma (DLBCL)	113	113	100.0	0.0	0.0	0.0
	Follicular lymphoma	88	88	100.0	0.0	0.0	0.0
	T-cell non-Hodgkin’s lymphoma	25	25	100.0	0.0	0.0	0.0
	Mantle cell lymphoma	18	18	100.0	0.0	0.0	0.0
	Marginal zone lymphoma	16	16	100.0	0.0	0.0	0.0
	Diffuse large B-cell lymphoma (DLBCL) in the testis	16	16	100.0	0.0	0.0	0.0
	Burkitt lymphoma	5	5	100.0	0.0	0.0	0.0
Tumors of soft tissue and bone	Granular cell tumor	23	21	76.2	9.5	14.3	0.0
	Leiomyoma	50	45	97.8	2.2	0.0	0.0
	Leiomyosarcoma	94	88	100.0	0.0	0.0	0.0
	Liposarcoma	96	93	100.0	0.0	0.0	0.0
	Malignant peripheral nerve sheath tumor (MPNST)	15	13	100.0	0.0	0.0	0.0
	Myofibrosarcoma	26	26	100.0	0.0	0.0	0.0
	Angiosarcoma	42	40	100.0	0.0	0.0	0.0
	Angiomyolipoma	91	89	100.0	0.0	0.0	0.0
	Dermatofibrosarcoma protuberans	21	17	100.0	0.0	0.0	0.0
	Ganglioneuroma	14	12	100.0	0.0	0.0	0.0
	Kaposi sarcoma	8	6	100.0	0.0	0.0	0.0
	Neurofibroma	117	116	100.0	0.0	0.0	0.0
	Sarcoma, not otherwise specified (NOS)	74	72	100.0	0.0	0.0	0.0
	Paraganglioma	41	40	100.0	0.0	0.0	0.0
	Ewing sarcoma	23	18	100.0	0.0	0.0	0.0
	Rhabdomyosarcoma	7	7	100.0	0.0	0.0	0.0
	Schwannoma	122	119	99.2	0.8	0.0	0.0
	Synovial sarcoma	12	11	100.0	0.0	0.0	0.0
	Osteosarcoma	19	18	100.0	0.0	0.0	0.0
	Chondrosarcoma	15	11	100.0	0.0	0.0	0.0
	Rhabdoid tumor	5	5	100.0	0.0	0.0	0.0
	Solitary fibrous tumor	17	14	100.0	0.0	0.0	0.0

**Table 2 diagnostics-14-02157-t002:** TFF1 immunostaining and tumor phenotype.

			TFF1 Immunostaining					
		n	Negative (%)	Weak (%)	Moderate (%)	Strong (%)	*p*	Remarks
Invasive breast carcinoma of no special type	pT1	802	448 (55.9)	203 (25.3)	94 (11.7)	57 (7.1)	0.3766	
	pT2	635	331 (52.1)	185 (29.1)	67 (10.6)	52 (8.2)		
	pT3-4	126	71 (56.3)	35 (27.8)	15 (11.9)	5 (4)		
	G1	202	106 (52.5)	52 (25.7)	32 (15.8)	12 (5.9)	<0.0001	
	G2	845	423 (50.1)	238 (28.2)	110 (13)	74 (8.8)		
	G3	564	354 (62.8)	146 (25.9)	36 (6.4)	28 (5)		
	pN0	692	361 (52.2)	174 (25.1)	102 (14.7)	55 (8)	0.3307	
	pN+	946	521 (55.1)	241 (25.5)	111 (11.7)	73 (7.7)		
	pM0	217	121 (55.8)	52 (24)	28 (12.9)	16 (7.4)	0.7665	
	pM1	112	60 (53.6)	32 (28.6)	14 (12.5)	6 (5.4)		
	HER2 negative	890	479 (53.8)	258 (29)	97 (10.9)	56 (6.3)	0.9505	
	HER2 positive	122	69 (56.6)	33 (27.1)	13 (10.7)	7 (5.7)		
	ER negative	213	184 (86.4)	21 (9.9)	3 (1.4)	5 (2.4)	<0.0001	
	ER positive	745	334 (44.8)	251 (33.7)	105 (14.1)	55 (7.4)		
	PR negative	408	277 (67.9)	90 (22.1)	27 (6.6)	14 (3.4)	<0.0001	
	PR positive	598	270 (45.2)	197 (32.9)	82 (13.7)	49 (8.2)		
	non-triple negative	783	366 (46.7)	258 (33)	102 (13)	57 (7.3)	<0.0001	
	triple negative	144	133 (92.4)	7 (4.9)	1 (0.7)	3 (2.1)		
Urothelial bladder carcinoma	pTa G2 low	152	115 (75.7)	30 (19.7)	5 (3.3)	2 (1.3)	<0.0001	
	pTa G2 high	116	96 (82.8)	16 (13.8)	3 (2.6)	1 (0.9)	0.0072	1
	pTa G3	99	91 (91.9)	5 (5.1)	3 (3)	0 (0)		
	pT2-4	453	429 (94.7)	19 (4.2)	5 (1.1)	0 (0)		
	pT2	125	122 (97.6)	3 (2.4)	0 (0)	0 (0)	0.3037	2,3,4
	pT3	219	206 (94.1)	9 (4.1)	4 (1.8)	0 (0)		
	pT4	98	91 (92.9)	6 (6.1)	1 (1)	0 (0)		
	G2	23	20 (87)	3 (13)	0 (0)	0 (0)	0.0995	2,4
	G3	429	408 (95.1)	16 (3.7)	5 (1.2)	0 (0)		
	pN0	271	261 (96.3)	8 (3)	2 (0.7)	0 (0)	0.0414	2,3
	pN+	161	148 (91.9)	11 (6.8)	2 (1.2)	0 (0)		
Adenocarcinoma of the pancreas	pT1	14	3 (21.4)	6 (42.9)	2 (14.3)	3 (21.4)	0.3769	5
	pT2	66	18 (27.3)	22 (33.3)	15 (22.7)	11 (16.7)		
	pT3	378	133 (35.2)	125 (33.1)	58 (15.3)	62 (16.4)		
	pT4	27	12 (44.4)	5 (18.5)	8 (29.6)	2 (7.4)		
	G1	17	5 (29.4)	7 (41.2)	2 (11.8)	3 (17.6)	0.8595	6
	G2	343	115 (33.5)	111 (32.4)	61 (17.8)	56 (16.3)		
	G3	103	39 (37.9)	32 (31.1)	17 (16.5)	15 (14.6)		
	pN0	101	28 (27.7)	34 (33.7)	18 (17.8)	21 (20.8)	0.3281	
	pN+	383	137 (35.8)	125 (32.6)	65 (17)	56 (14.6)		
	R0	244	80 (32.8)	90 (36.9)	39 (16)	35 (14.3)	0.4142	
	R1	201	72 (35.8)	59 (29.4)	37 (18.4)	33 (16.4)		
Adenocarcinoma of the stomach	pT1-2	62	31 (50)	21 (33.9)	7 (11.3)	3 (4.8)	0.8258	3
	pT3	128	70 (54.7)	28 (21.9)	13 (10.2)	17 (13.3)		
	pT4	122	64 (52.5)	26 (21.3)	14 (11.5)	18 (14.8)		
	pN0	85	42 (49.4)	23 (27.1)	10 (11.8)	10 (11.8)		
	pN+	226	123 (54.4)	50 (22.1)	24 (10.6)	29 (12.8)		
	MMR proficient	257	136 (52.9)	60 (23.3)	34 (13.2)	27 (10.5)	0.6225	3
	MMR deficient	41	20 (48.8)	14 (34.1)	1 (2.4)	6 (14.6)		
Adenocarcinoma of the colon	pT1	75	30 (40)	39 (52)	4 (5.3)	2 (2.7)	0.6503	5
	pT2	421	220 (52.3)	141 (33.5)	39 (9.3)	21 (5)		
	pT3	1200	604 (50.3)	455 (37.9)	86 (7.2)	55 (4.6)		
	pT4	435	227 (52.2)	153 (35.2)	28 (6.4)	27 (6.2)		
	pN0	1125	575 (51.1)	413 (36.7)	82 (7.3)	55 (4.9)	0.9973	
	pN+	1003	516 (51.4)	364 (36.3)	73 (7.3)	50 (5)		
	V0	1533	783 (51.1)	555 (36.2)	112 (7.3)	81 (5.3)	0.6924	
	V1	563	291 (51.7)	208 (36.9)	41 (7.3)	23 (4.1)		
	L0	701	369 (52.6)	240 (34.2)	54 (7.7)	38 (5.4)	0.4669	
	L1	1404	706 (50.3)	529 (37.7)	102 (7.3)	67 (4.8)		
	right side	427	188 (44)	168 (39.3)	47 (11)	24 (5.6)	0.0021	
	left side	1173	622 (53)	416 (35.5)	75 (6.4)	59 (5)		
	MMR proficient	1110	586 (52.8)	402 (36.2)	75 (6.8)	47 (4.2)	0.0292	3
	MMR deficient	84	34 (40.5)	35 (41.7)	11 (13.1)	4 (4.8)		
	RAS wildtype	450	274 (60.9)	131 (29.1)	23 (5.1)	22 (4.9)	<0.0001	
	RAS mutation	340	148 (43.5)	138 (40.6)	38 (11.2)	16 (4.7)		
	BRAF wildtype	111	70 (63.1)	30 (27)	8 (7.2)	3 (2.7)	0.0331	3
	BRAF V600E mutation	19	7 (36.8)	6 (31.6)	3 (15.8)	3 (15.8)		

Abbreviations: pT: pathological tumor stage, G: Grade, pN: pathological lymph node status, pM: pathological status of distant metastasis, L: lymphatic infiltration, V: venous infiltration, R: resection margin status, MMR: mismatch repair. Remarks: *p*-values corespond to 1: pTa only, 2: pT2-4 only, 3: TFF1 negative vs. TFF1 positive (weak, mod. str.), 4: Fisher’s excact test, 5: pT1-2 vs. pT3-4, 6: G1-2 vs. G3.

## Data Availability

Dataset available on request from the authors.

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
