# Peer review of "Expression of Trefoil Factor 1 (TFF1) in Cancer: A Tissue Microarray Study Involving 18,878 Tumors"

_diagnostics, 2024, doi:10.3390/diagnostics14192157_

Round 1

Reviewer 1 Report

Comments and Suggestions for Authors

The authors analyzed the expression of TFF1 in various tumors by tissue microarray. The results are well-presented so that we can understand that TFF1 is expressed in a certain type of cancer, including ovarian, colorectal, breast, gastrointestinal, cervical, and urothelial tumors. Although these data are informative to a certain extent, some concerns are listed below.

1.       I agree that “TFF1 upregulation is a common phenomenon in cancer which may have a significant functional role”. Then, what is the role of TFF1? Tumor suppressor or oncogenic effector? Authors should at least consider the role of TFF1 in malignant tumors based on the findings described in the results.

2.       The findings of predominant TFF1 expression in low-grade malignancies is interesting. Given that adenomatous polyp express TFF1, it is likely that premalignant lesions, rather than carcinoma cells, express TFF1. Authors had better analyze other premalignant lesions such as gastric adenoma, pancreatic intraepithelial neoplasia, prostatic intraepithelial neoplasia, atypical ductal hyperplasia of the breast, cervical intraepithelial neoplasia, etc. This would improve the significance of this experiments.  

3.       Authors claimed that “striking discrepancy between … normal tissues and …cancer types”, but there is no comparison of the data, except for a few IHC images in normal tissues. The difference should be quantified and analyzed.

4.       As authors say, TFF1 is not useful to discriminate the different tumor entities. Then, how can we utilize the results of this experiment concerning TFF1 expression? The early detection of cancerous lesions? For precision medicine? Please exhibit potential benefits and future perspectives.

Author Response

In reply to the comments and suggestions of reviewer 1, we made the following changes:

Reviewer 1:

The authors analyzed the expression of TFF1 in various tumors by tissue microarray. The results are well-presented so that we can understand that TFF1 is expressed in a certain type of cancer, including ovarian, colorectal, breast, gastrointestinal, cervical, and urothelial tumors. Although these data are informative to a certain extent, some concerns are listed below.

1. I agree that “TFF1 upregulation is a common phenomenon in cancer which may have a significant functional role”. Then, what is the role of TFF1? Tumor suppressor or oncogenic effector? Authors should at least consider the role of TFF1 in malignant tumors based on the findings described in the results. 

Reply: We have now expanded our discussion on the functional consequences of TFF1 deregulation on page 14, lines 240-244.

  1. The findings of predominant TFF1 expression in low-grade malignancies is interesting. Given that adenomatous polyp express TFF1, it is likely that premalignant lesions, rather than carcinoma cells, express TFF1. Authors had better analyze other premalignant lesions such as gastric adenoma, pancreatic intraepithelial neoplasia, prostatic intraepithelial neoplasia, atypical ductal hyperplasia of the breast, cervical intraepithelial neoplasia, etc. This would improve the significance of this experiments. 

Reply: We have now commented on TFF1 expression in premalignant lesion on page 14, lines 232 - 234.  

3.       Authors claimed that “striking discrepancy between … normal tissues and …cancer types”, but there is no comparison of the data, except for a few IHC images in normal tissues. The difference should be quantified and analyzed.

Reply: We have now added supplementary table 1 comparing TFF1 staining in normal and cancer tissues.

  1. As authors say, TFF1 is not useful to discriminate the different tumor entities. Then, how can we utilize the results of this experiment concerning TFF1 expression? The early detection of cancerous lesions? For precision medicine? Please exhibit potential benefits and future perspectives.

Reply: We have now expanded our para on the future use and perspectives of TFF1 IHC analysis on page 13, lines 216-219.

We thank the reviewer for his valuable comments and suggestions, and the time devoted to our manuscript.

Sincerely,

Ronald Simon

Reviewer 2 Report

Comments and Suggestions for Authors

Regarding the statistical analysis in Table 2, there are a few key issues that need to be clarified to enhance the credibility of the analysis results.

1. Table Formatting: It is recommended to add horizontal lines to the table to clearly separate the results of each chi-square test from the contingency tables.

2. Contingency Table Presentation: Typically, contingency tables used and recorded for chi-square tests display counts rather than just percentages. It is advisable to present both counts and percentages together.

3. Chi-Square Test Assumption: The assumption of the chi-square test requires that each sample state belongs to only one cell in the contingency table. Please ensure that the phenotypes within each chi-square test are mutually exclusive.

4. Expected Cell Count: Another assumption of the chi-square test is that each cell in the contingency table should have an expected count of at least 5. Given that many cells in Table 2 have percentages of 0, the expected counts are also 0, potentially violating this assumption. It is recommended to use Fisher’s Exact Test or to combine categories to address this issue. Additionally, recording the degrees of freedom for the chi-square statistic in the table can ensure the correct application of the chi-square test.

Additionally, the tables in this article are all long tables without suitable separators. It is recommended to revise the table format to make it clearer for readers.

Author Response

Dear Prof Kjaer,

In reply to the comments and suggestions of reviewer 2, we made the following changes:

Reviewer 2:

Regarding the statistical analysis in Table 2, there are a few key issues that need to be clarified to enhance the credibility of the analysis results.

  1. Table Formatting: It is recommended to add horizontal lines to the table to clearly separate the results of each chi-square test from the contingency tables.

Reply: Following the reviewer’s suggestion, we have added horizontal lines to the table.

  1. Contingency Table Presentation: Typically, contingency tables used and recorded for chi-square tests display counts rather than just percentages. It is advisable to present both counts and percentages together.

Reply: Following the reviewer’s suggestion, we have added counts to the table.

  1. Chi-Square Test Assumption: The assumption of the chi-square test requires that each sample state belongs to only one cell in the contingency table. Please ensure that the phenotypes within each chi-square test are mutually exclusive.

Reply: We confirm that the samples in each category/phenotype are mutually exclusive.

  1. Expected Cell Count: Another assumption of the chi-square test is that each cell in the contingency table should have an expected count of at least 5. Given that many cells in Table 2 have percentages of 0, the expected counts are also 0, potentially violating this assumption. It is recommended to use Fisher’s Exact Test or to combine categories to address this issue. Additionally, recording the degrees of freedom for the chi-square statistic in the table can ensure the correct application of the chi-square test.

Reply: We have combined categories and/or used Fisher’s exact test in cases with counts ≤5 in table 2, defined these analyses in the table footer, and added this information to the methods section.

Additionally, the tables in this article are all long tables without suitable separators. It is recommended to revise the table format to make it clearer for readers.

Reply: We have now better structured the long tables using horizontal lines to increase readability.

We thank the reviewer for his valuable comments and suggestions, and the time devoted to our manuscript.

Sincerely,

Ronald Simon

Round 2

Reviewer 1 Report

Comments and Suggestions for Authors

The authors responded to all of the comments.

Reviewer 2 Report

Comments and Suggestions for Authors

In this version, the authors have made appropriate revisions based on the reviewers' suggestions, and therefore, no further recommendations are necessary.